# Design of the Drive Mechanism of a Rotating Feeding Device

Matteo Bottin *, Riccardo Minto and Giulio Rosati

Department of Industrial Engineering, University of Padova, 35131 Padova, Italy
* Correspondence: matteo.bottin@unipd.it

**Abstract:** Component batching can be a source of time waste in specific industrial applications, such as kitting. Kitting operations are usually performed by hoppers, but other devices can be used to optimize the process. In a previous work, a rotary device has been proved to be more efficient than hoppers; such a device allows the kitting and releasing of the components in a single rotatory movement, while traditional hoppers require at least two movements. In this paper, an improvement of such feeding device is proposed. The movement of the rotary device is driven by a four-bar linkage mechanism which is designed through functional synthesis. Thank to the four-bar linkage mechanism, the alternate motion of the rotary distributor is derived from the constant speed of the motor.

**Keywords:** feeding; rotary device; functional synthesis; mechanical design

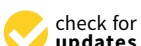



## 1. Introduction

The target of many companies is to either reduce production costs or increase the throughput. This is important, especially for Western countries, where the unit direct production cost of goods in the manufacturing industry is driven mainly by the cost of labor. To overcome such issue, production systems must be optimized by increasing their flexibility [1,2].

In the manufacturing industry, kitting lines play a major role. The aim of kitting lines is to create kits of different objects with predetermined quantities [3]. To achieve this, firstly, small batches are created, one for each type of object; then, the batches are grouped together to assemble the kit. Such kits may be sold to the final customer, called *sales kit*, or prepared for other assembly processes, called *production kit*. To create the batches, a common solution is given by hopper systems. Hoppers are used in series to group components, perform quality checks through weight measurements, and act as buffers. However, hoppers suffer from a non-negligible drawback, i.e., for each release operation, they need two movements, opening, and closing. To overcome such issue, hopper design has been studied in the last years, and both axial-symmetric [4–8] and eccentric [9,10] solutions have been analyzed, trying to optimize the flow of the components to reduce the drawback. However, hoppers are yet a source of inefficiency [11]. For small kit sizes, robots can be used as an alternate means of kitting [12,13]; however, robot productivity is too low with respect to hoppers when the number of parts per kit is large.

To the best of the authors' knowledge, there is a general lack of alternatives to hoppers; indeed, very few works have been published in this area in the last few years. For example, Pantyukhina et al. [14] proposed a mechanical toothed hopper-feeding device whose purpose is not component kitting, but component orientation. Gao et al. [15] studied a rotational device to be used for a continuous flow of small particles; thus, the purpose and design are different from a kitting operation, in which rather large components are kitted.

In this paper, a novel solution for component batching is proposed by means of a compartmentalized rotating device. Such a device, which follows the concepts introduced in previous works [16,17], requires only a single rotational movement for each release operation. In fact, it is made up of compartments divided by blades specifically

designed for optimized component falling. This work focuses both on the mechanical design and how it affects the blade shape to ensure the free-falling motion of the components. In particular, we propose to use a four-bar linkage mechanism to drive the movement of the blades; in this way, an alternate motion of the blades is obtained without requiring complex electronics, since it is derived from the motion of an electric motor driven at a constant speed. This design choice on the one side simplifies the control system of the rotating device, which is the main advantage provided by the solution proposed, on the other side it requires a re-design of the blades. Both aspects are addressed in the paper.

The work is structured as follows. Firstly, the mechanical design of both the mechanism and the blades is presented in Section 2. Then, the numerical results of the design principles are presented in Section 3. Finally, experimental tests are described in Section 4, and the performances are evaluated.

## 2. Mechanical Design

The proposed device is shown in Figure 1. The device is composed of three main parts:

- A fixed cylindrical structure, in brown in Figure 1a, which acts as a container for the components to be kitted. The cylinder axis is horizontal, and the curved surface has two openings: one at the top to receive new components, and one at the bottom to exit the components.
- The rotary distributor, in teal green in Figure 1a, coaxial with the cylinder, which divides the cylinder into two compartments via some blades. The rotation of the blades around the cylinder axis allows the components to fall within one compartment or the other (Figure 2).
- A four-bar linkage mechanism, in light grey in Figure 1a, whose rocker link is fixed with the blades (angle $\varphi_3$ of Figure 3). The crank link is controlled by means of an electric motor that rotates at constant speed, whose absolute angle is $q$.

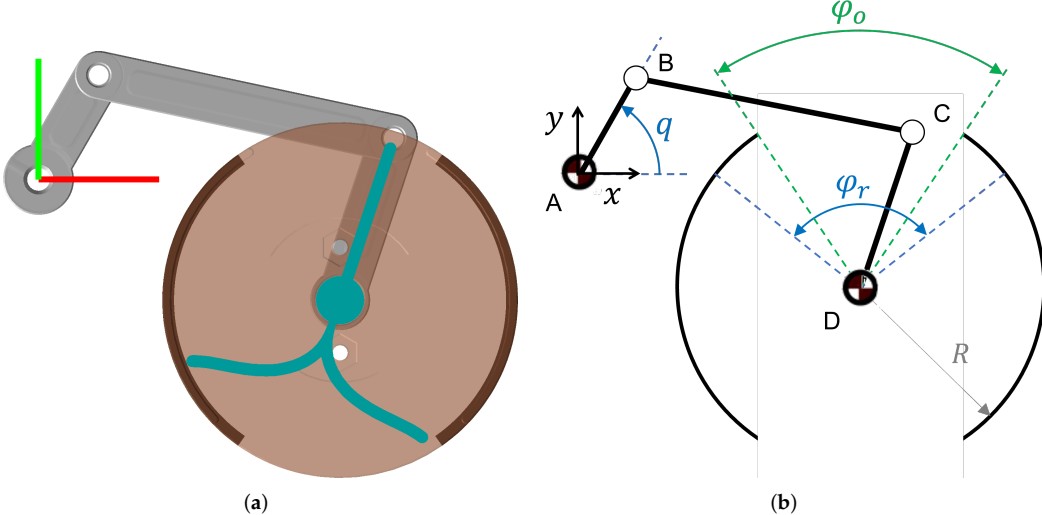

|        (**a**)        |        (**b**)        |

**Figure 1.** Design of the proposed mechanism. CAD model (**a**) with the four-bar linkage system (light grey) and the rotary distributor with the blades (teal green); mathematical model (**b**).

The four-bar linkage mechanism has been chosen for its extreme simplicity both in terms of control, manufacturing, and costs. In fact, the device can be built by means of cheap 3D rapid prototyping with great results.

Exploiting the movement of the mechanism, if the crank link rotates with a fixed speed the blades rotate periodically to the left and to the right, allowing the components to fall within one compartment or the other (an example can be seen in Figure 2). This allows the machine to be controlled by a very simple control system.

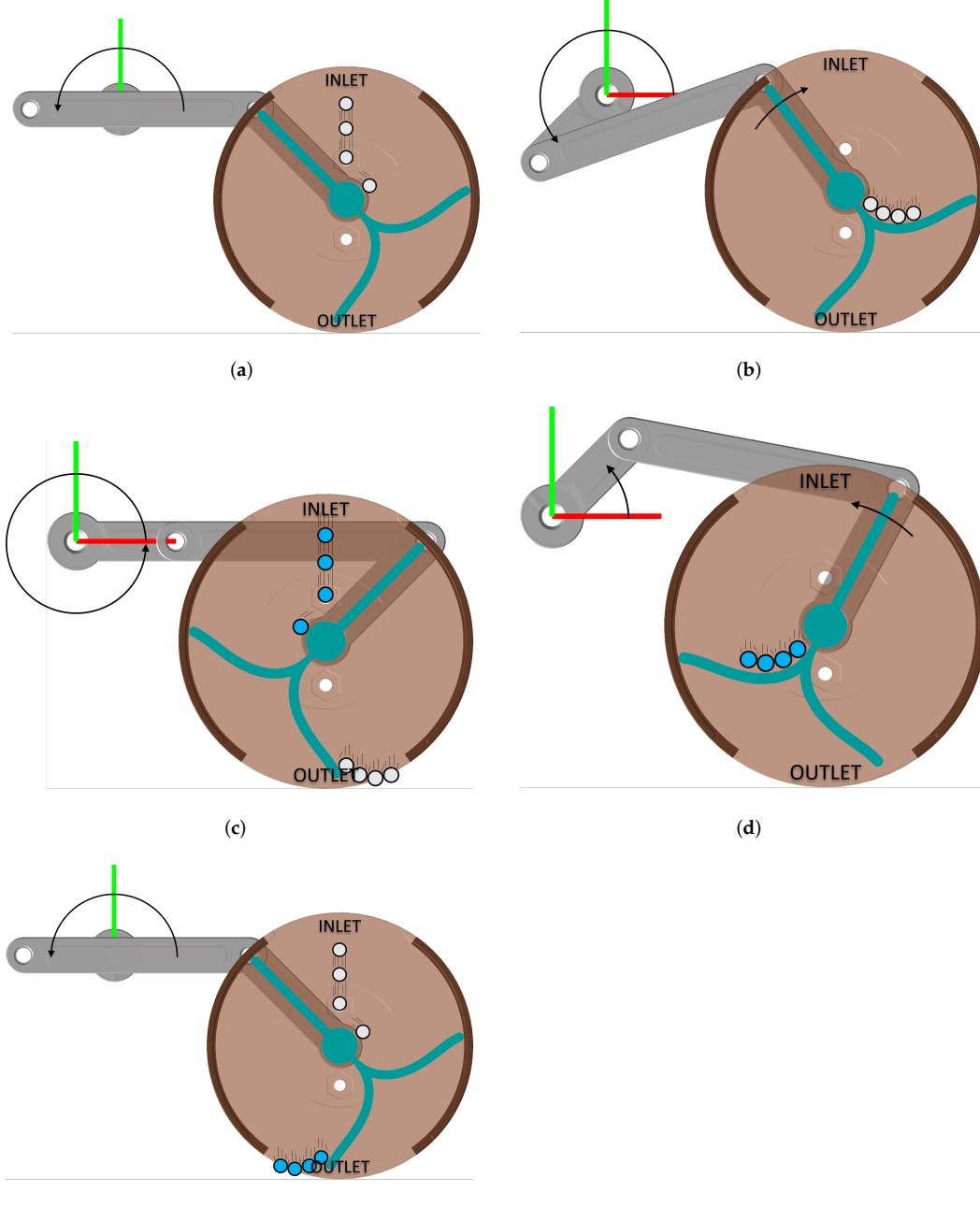

**Figure 2.** Functioning of the mechanism. The rotary distributor (teal green) moves to the left and to the right to batch components, while the four-bar linkage mechanism is driven by a motor fixed to the crank which rotates at constant speed. The parts fall into the compartment from the inlet (to the top) and exit from the outlet (to the bottom) under the action of gravity. In the sequence depicted above, the first set of parts (white) exits to the right (**a–c**), and the second set (blue) exits to the left (**c–e**).

## 2.1. Design of the Four-Bar Linkage Mechanism

The design of the four-bar linkage can be performed via precision-point synthesis [18]. Precision-point synthesis allows sizing the mechanism so that it crosses specific configurations, called precision points. Such configurations are chosen by design to exploit specific mechanism behavior.

Let us consider the vector scheme of Figure 3. The position closure equation can be written as follows:

$$\sum_{i=\{1,...,4\}} \pm \mathbf{z}_i = \mathbf{0} \quad \rightarrow \quad \mathbf{z}_1 + \mathbf{z}_2 - \mathbf{z}_3 - \mathbf{z}_4 = \mathbf{0} \tag{1}$$

which must hold for every mechanism configuration. If a configuration 0 is considered, each vector $i$ complex form is:

$$\mathbf{z}_{i,0} = a_i e^{j\varphi_{i,0}} \tag{2}$$

where $a_i$ is the vector length and $\varphi_{i,0}$ is the absolute angle. If a rotation $\delta_{i,k}$ is applied to vector $\mathbf{z}_{i,0}$, the resulting vector $\mathbf{z}_{i,k}$ is:

$$\mathbf{z}_{i,k} = a_i e^{j(\varphi_{i,0}+\delta_{i,k})} = \mathbf{z}_{i,0} e^{j\delta_{i,k}} \tag{3}$$

which leads to the position closure equation for configuration $k$:

$$\sum_{i=\{1,...,4\}} \pm \mathbf{z}_{i,k} = \sum_{i=\{1,...,4\}} \pm \mathbf{z}_{i,0} e^{j\delta_{i,k}} = \mathbf{0} \tag{4}$$

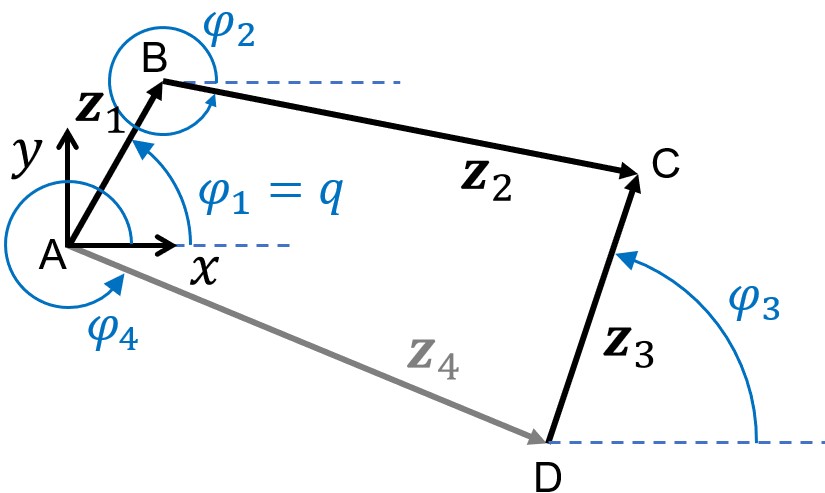

**Figure 3.** Schematic of the four-bar linkage mechanism. The vectors represent the position of the links of the mechanism.

Subtracting Equations (1) and (4) we get:

$$\sum_{i=\{1,...,4\}} \pm (\mathbf{z}_{i,k} - \mathbf{z}_{i,0}) = \sum_{i=\{1,...,4\}} \pm \mathbf{z}_{i,0} (e^{j\delta_{i,k}} - 1) = \mathbf{0} \tag{5}$$

which represents the *position closure equation of the displacements*. For displacement $k$, Equation (5) becomes:

$$\mathbf{z}_{1,0}(e^{j\delta_{1,k}} - 1) + \mathbf{z}_{2,0}(e^{j\delta_{2,k}} - 1) - \mathbf{z}_{3,0}(e^{j\delta_{3,k}} - 1) = \mathbf{0} \tag{6}$$

where $\mathbf{z}_{4,0}$ has been deleted since it is fixed, i.e., $\mathbf{z}_{4,k} = \mathbf{z}_{4,0}$; thus, $\delta_{4,k} = 0$ for every displacement $k$.

Equation (6) can be used to size the mechanism based on the design requirements. In particular, the designer is usually interested in designing the mechanism such as that a specific displacement of the crank $\delta_{1,k}$ corresponds to a specific displacement of the rocker $\delta_{3,k}$ (the so-called *functional synthesis*). As a result, the number of unknowns in Equation (6) is 7: the three vector lengths ($a_1$, $a_2$, $a_3$) and orientations ($\varphi_{1,0}$, $\varphi_{2,0}$, $\varphi_{3,0}$),

and the displacement of the coupler link ($\delta_{2,k}$). The corresponding number of scalar equations is 2, one for the real part and one for the imaginary part of the vectorial equation. Vector $\mathbf{z}_4$ length ($a_4$) and orientation ($\varphi_{4,0}$) are calculated by means of Equation (1) for configuration 0.

As a result, configuration 0 is not considered—since its equations are used to calculate only $\mathbf{z}_4$—thus, the number of unknowns for $N$ displacements is $6 + N$, whereas the number of equations is $2N$. Table 1 shows the number of unknown parameters for $N$ displacements up to 6. One of the main advantages of the functional synthesis of a mechanism is that the solution to be found can be scaled and oriented according to the needs. That said, if a length of a link is fixed, all the other links can be calculated based on such length; if the length is doubled, all the other links are doubled as well. Moreover, the same mechanism can be rotated while maintaining the functionality [19].

**Table 1.** Number of unknowns, equations and solutions for $N$ displacements.

| $N$ | Unknowns | Equations | Solutions |
|---|---|---|---|
| 1 | 7 ($a_1,a_2,a_3,\varphi_{1,0},\varphi_{2,0},\varphi_{3,0},\delta_{2,1}$) | 2 | $\infty^5$ |
| 2 | 8 ($a_1,a_2,a_3,\varphi_{1,0},\varphi_{2,0},\varphi_{3,0},\delta_{2,1},\delta_{2,2}$) | 4 | $\infty^4$ |
| 3 | 9 ($a_1,a_2,a_3,\varphi_{1,0},\varphi_{2,0},\varphi_{3,0},\delta_{2,1},\delta_{2,2},\delta_{2,3}$) | 6 | $\infty^3$ |
| 4 | 10 ($a_1,a_2,a_3,\varphi_{1,0},\varphi_{2,0},\varphi_{3,0},\delta_{2,1},\delta_{2,2},\delta_{2,3},\delta_{2,4}$) | 8 | $\infty^2$ |
| 5 | 11 ($a_1,a_2,a_3,\varphi_{1,0},\varphi_{2,0},\varphi_{3,0},\delta_{2,1},\delta_{2,2},\delta_{2,3},\delta_{2,4},\delta_{2,5}$) | 10 | $\infty^1$ |
| 6 | 12 ($a_1,a_2,a_3,\varphi_{1,0},\varphi_{2,0},\varphi_{3,0},\delta_{2,1},\delta_{2,2},\delta_{2,3},\delta_{2,4},\delta_{2,5},\delta_{2,6}$) | 12 | 1 |

If $N = 3$, the corresponding equations retrieved from Equation (6) can be compacted in a matrix form as follows:

$$\mathbf{E} \cdot \mathbf{z}_0 = \begin{bmatrix} e^{j\delta_{1,1}} - 1 & e^{j\delta_{2,1}} - 1 & 1 - e^{j\delta_{3,1}} \\ e^{j\delta_{1,2}} - 1 & e^{j\delta_{2,2}} - 1 & 1 - e^{j\delta_{3,2}} \\ e^{j\delta_{1,3}} - 1 & e^{j\delta_{2,3}} - 1 & 1 - e^{j\delta_{3,3}} \end{bmatrix} \begin{Bmatrix} \mathbf{z}_{1,0} \\ \mathbf{z}_{2,0} \\ \mathbf{z}_{3,0} \end{Bmatrix} = \begin{Bmatrix} \mathbf{0} \\ \mathbf{0} \\ \mathbf{0} \end{Bmatrix} \tag{7}$$

where, as stated previously, $\mathbf{z}_{1,0}$ and $\mathbf{z}_{3,0}$ displacements ($\delta_{1,k}, \delta_{3,k}$) are known by design. Systems with $N > 3$ are more difficult to solve due to the coupling of the parameters and the fact that matrix $\mathbf{E}$ is non-squared, but some methods have been proposed [20–22].

From Table 1, it can be inferred that there are $\infty^3$ solutions to the system of Equation (7). As a result, three parameters must be chosen. The unknowns are the three vectors and the displacements of the coupler link ($\delta_{2,1},\delta_{2,2},\delta_{2,3}$). The algebraic system admits solutions different from the trivial one if and only if the determinant of the matrix of displacements $\mathbf{E}$ is equal to zero. In other words, it is possible to choose one displacement of the coupler ($\delta_{2,k}$) to calculate the others. Then, the system can be solved by choosing the length and the orientation of one of the vectors. Finally, $\mathbf{z}_4$ is calculated by means of (1) for configuration 0. Please note that by choosing one vector and one displacement, the number of unknowns reduces by 3; thus, the solution is unique.

The determinant of the matrix of displacements is:

$$\det\left( \begin{bmatrix} e^{j\delta_{1,1}} - 1 & e^{j\delta_{2,1}} - 1 & 1 - e^{j\delta_{3,1}} \\ e^{j\delta_{1,2}} - 1 & e^{j\delta_{2,2}} - 1 & 1 - e^{j\delta_{3,2}} \\ e^{j\delta_{1,3}} - 1 & e^{j\delta_{2,3}} - 1 & 1 - e^{j\delta_{3,3}} \end{bmatrix} \right) = \mathbf{E}_{21}e^{j\delta_{2,1}} + \mathbf{E}_{22}e^{j\delta_{2,2}} + \mathbf{E}_{23}e^{j\delta_{2,3}} + \mathbf{E}_0 = \mathbf{0} \tag{8}$$

where

$$\mathbf{E}_{21} = e^{j\delta_{1,3}} - e^{j\delta_{1,2}} + e^{j\delta_{3,2}} - e^{j\delta_{3,3}} + e^{j\delta_{1,2}}e^{j\delta_{3,3}} - e^{j\delta_{1,3}}e^{j\delta_{3,2}} \tag{9}$$

$$\mathbf{E}_{22} = e^{j\delta_{1,1}} - e^{j\delta_{1,3}} - e^{j\delta_{3,1}} + e^{j\delta_{3,3}} - e^{j\delta_{1,1}}e^{j\delta_{3,3}} + e^{j\delta_{1,3}}e^{j\delta_{3,1}} \tag{10}$$

$$\mathbf{E}_{23} = e^{j\delta_{1,2}} - e^{j\delta_{1,1}} + e^{j\delta_{3,1}} - e^{j\delta_{3,2}} + e^{j\delta_{1,1}}e^{j\delta_{3,2}} - e^{j\delta_{1,2}}e^{j\delta_{3,1}} \tag{11}$$

$$\mathbf{E}_0 = e^{j\delta_{1,2}}e^{j\delta_{3,1}} - e^{j\delta_{1,1}}e^{j\delta_{3,2}} + e^{j\delta_{1,1}}e^{j\delta_{3,3}} - e^{j\delta_{1,3}}e^{j\delta_{3,1}} - e^{j\delta_{1,2}}e^{j\delta_{3,3}} - e^{j\delta_{1,3}}e^{j\delta_{3,2}} \tag{12}$$

where all the terms are known since are design variables.

Equation (8) is a non-linear complex equation system that can be solved numerically by choosing one of the displacements $\delta_{2,k}$. In fact, the system has two equations (one for the real part and the other for the imaginary part) but three unknowns ($\delta_{2,1}, \delta_{2,2}, \delta_{2,3}$). By choosing one of such unknowns it is possible to solve the system.

Once the displacements are found, the system of Equation (7) can be solved by removing one row (since the determinant of the $3 \times 3$ matrix is null). The resulting system is a $2 \times 2$ system with 3 unknown vectors. As a result, it is necessary to choose one of such vectors to solve the system. Without losing in generality, lets us consider that vector $\mathbf{z}_{3,0}$ has been chosen and the third equation removed:

$$
\begin{bmatrix} e^{j\delta_{1,1}} - 1 & e^{j\delta_{2,1}} - 1 \\ e^{j\delta_{1,2}} - 1 & e^{j\delta_{2,2}} - 1 \end{bmatrix} \begin{Bmatrix} \mathbf{z}_{1,0} \\ \mathbf{z}_{2,0} \end{Bmatrix} = - \begin{Bmatrix} 1 - e^{j\delta_{3,1}} \\ 1 - e^{j\delta_{3,2}} \end{Bmatrix} \mathbf{z}_{3,0}
\tag{13}
$$

which can be solved if the rank of the matrix of displacements is equal to 2. From Equation (13) the three vectors $\mathbf{z}_{1,0}$, $\mathbf{z}_{2,0}$, and $\mathbf{z}_{3,0}$ are fully defined, while $\mathbf{z}_{4,0}$ is calculated by means of Equation (1) for configuration 0:

$$
\mathbf{z}_{4,0} = \mathbf{z}_{1,0} + \mathbf{z}_{2,0} - \mathbf{z}_{3,0}
\tag{14}
$$

Finally, link length $a_i$ and orientation $\varphi_{i,0}$ are calculated from the complex vector form:

$$
\begin{aligned}
a_i &= |\mathbf{z}_i| = \sqrt{\Re(\mathbf{z}_i)^2 + \Im(\mathbf{z}_i)^2} \\
\varphi_{i,0} &= \operatorname{atan2}(\Im(\mathbf{z}_i), \Re(\mathbf{z}_i))
\end{aligned}
\tag{15}
$$

where $\Re(\mathbf{z}_i)$ and $\Im(\mathbf{z}_i)$ are the real and imaginary part of the complex vector $\mathbf{z}_i$.

It is worth noting that the solution given from the functional synthesis allows the mechanism to fulfill the required configurations, but nothing can be said about the behavior of the mechanism in all the other configurations. In other words, it is possible that the crank is not able to perform a complete rotation around its axis. Further checks must be performed to address the validity of the solution based on specific design requirements, such as the Grashof Law. Finally, $\mathbf{z}_{3,0}$ has to be chosen in such a way that even the largest parts to be kitted can easily fall within the compartments. In this sense, the mechanism and the motor must be scaled accordingly.

Provided that the actuation torque of a four-bar linkage mechanism may be very high when a singular configuration is approached, the mechanism obtained by the synthesis must be checked to verify that the actuation torque is bounded within reasonable values. If the solution is not applicable, the design must restart and different values on the constraints (i.e., displacements $\delta_{1,k}, \delta_{2,k}, \delta_{3,k}$ and one vector) must be applied.

### 2.2. Design of the Blades

In a hopper system, the easiest trajectory a component should follow is the free falling motion, without any interaction with the blade [16]. Indeed, this is the trajectory that requires the smallest time to let the component exit the cylinder volume. Different component behavior could be exploited by means of more complex interaction models, such as by considering the dynamics of the components [17]. The optimal design of the blade is the one that follows the movement of the component during the free fall without interacting with it.

The steps for the design of the blades are [16]:

1. The starting position of the component falling on the blade $(x_0, y_0)$ is calculated. This position is arbitrary and can be chosen according to specific needs (e.g., the motor shaft dimensions).

2. The free falling motion law of the component is calculated to retrieve the movement of the component in the absolute reference frame (0):

$$y(t) = y_0 - \frac{1}{2}gt^2 \tag{16}$$

3. By imposing a specific motion law of the rotary distributor it is possible to calculate the orientation of the reference frame relative to the rotary distributor. In particular, if $\psi(t)$ is the rotation angle of the rotary distributor, the transformation of coordinates between the absolute reference frame (0) and relative reference frame ($r$) is described by the transformation matrix:

$$\mathbf{T}_{r0}(t) = \begin{bmatrix} \cos(\psi(t)) & -\sin(\psi(t)) & 0 \\ \sin(\psi(t)) & \cos(\psi(t)) & 0 \\ 0 & 0 & 1 \end{bmatrix} \tag{17}$$

4. The position $\mathbf{P}_r(t)$ of the center of the component in the relative reference frame can be calculated by means of $T_{r0}$:

$$\mathbf{P}_r(t) = \left\{ \begin{matrix} x(t) \\ y(t) \\ 1 \end{matrix} \right\}_r = \mathbf{T}_{r0}(t) \left\{ \begin{matrix} x_0 \\ y(t) \\ 1 \end{matrix} \right\}_0 \tag{18}$$

5. The blade shape is the lower envelope of the circles centered in $\mathbf{P}_r(t)$ in the reference frame. In such a way, the blade will always be very close to the component during its fall.

From Equation (18) it can be inferred that the motion law of the rotary distributor $\psi(t)$ plays a major role in the design of the blade: different motion laws and rotational speeds result in very different blades. Particular attention must be paid to high-speed motion laws, since the incoming parts may rebound from the blades. However, it must be considered that the speed of the rocker (and of the blades) is very low near the dead points, right when the incoming parts enter and exit the device. In the case of a direct motor installed on the rotary distributor [16], $\psi(t)$ is described by well-known motion laws, such as third-degree polynomials or trapezoidal speed laws. Indeed, it is nearly impossible to drive the distributor at a constant speed, since it would allow a very short time for the components to fall within the compartments. Moreover, this type of control system requires control electronics which increases the costs of the device.

On the other hand, using a four-bar linkage allows the motor to be moved at a constant speed, while $\psi(t)$ is driven solely by the speed ratio between the crank and the rocker, which is given by the synthesis of the mechanism. In this case, $\psi(t) = \varphi_3(t)$, which can be derived directly by the kinematic of the four-bar linkage. In fact, the motion law can be expressed conveniently through its first derivative $\dot{\psi}(t)$:

$$\dot{\psi}(t) = \dot{\varphi}_3(t) = \frac{\mathrm{d}\varphi_3}{\mathrm{d}t} = \frac{\mathrm{d}\varphi_3}{\mathrm{d}q}\frac{\mathrm{d}q}{\mathrm{d}t} = w_{\varphi_3}(q)\dot{q}(t) \tag{19}$$

where $w_{\varphi_3}(q)$ is the speed ratio between the rocker and the crank and is:

$$w_{\varphi_3}(q) = \frac{a_1 \sin(\varphi_2 - q)}{a_3 \sin(\varphi_2 - \varphi_3)} \tag{20}$$

The motion law $\psi(t)$ can be derived via integration of Equation (19), where the differential variable is $q$, since in our case $\dot{q}$ is assumed to be constant.

## 3. Validation

### 3.1. Mechanism Synthesis

The methodology presented in the last section has been implemented to perform both the mechanism synthesis and the blade profile generation. For the mechanism synthesis, 3 precision points have been chosen. The precision points, which represent displacements from configuration 0, have been chosen to follow specific design principles:

- The angular top opening range $\varphi_O$ is arbitrary but should be wide enough to allow multiple pieces to fall within the device without falling outside of the cylinder. Moreover, it is mandatory that $\varphi_O \leq \varphi_r$. Indeed, if the top opening range is wider than the blade stroke, the two compartments will always have an open top gap which could allow components to pass through. As a result, the first configuration has been chosen so that $\delta_{3,1} = \varphi_r$. Since configuration 0 must be reached once every crank full rotation, this first displacement, which represents half of the full movement of the rocker, must be performed in half rotation; thus, $\delta_{1,1} = 180°$.
- The other two displacements are chosen so that the rocker moves from the dead points by a certain amount with the same crank displacement. In other words, $\delta_{3,3} = \delta_{3,1} - \delta_{3,2}$, and $\delta_{1,3} = 180° + \delta_{1,2}$.

Vector $\mathbf{z}_3$ has been chosen so that the rotary compartment fits the cylindrical part used in a previous work [16] and $\varphi_3 = \pi/4$. All the numerical values used in the synthesis are shown in Table 2. In our case $\varphi_r = 90°$, so for every crank full rotation the rocker must move 90° to the right and 90° to the left. It is worth noting that the synthesis does not ensure that the rocker moves exactly 90° every half crank rotation, but, thanks to $\delta_{3,1} = 90°$, ensures that the rocker moves at least 90° every half crank rotations.

**Table 2.** Numerical values used in the synthesis of the mechanism. Design requirements to the left, fixed unknown values to the right.

| Parameter | Value | Unit | Unknown | Value | Unit |
|-----------|-------|------|---------|-------|------|
| $\delta_{1,1}$ | 180 | [°] | $\delta_{2,3}$ | 20 | [°] |
| $\delta_{1,2}$ | 45 | [°] | $a_3$ | 56.6 | [mm] |
| $\delta_{1,3}$ | 225 | [°] | $\varphi_{3,0}$ | 45 | [°] |
| $\delta_{3,1}$ | 90 | [°] | | | |
| $\delta_{3,2}$ | 20 | [°] | | | |
| $\delta_{3,3}$ | 70 | [°] | | | |

Synthesis results are shown in Table 3 and in Figure 4. The configurations shown in Figure 4 repeat every crank rotation at specific $q$ values. The main result of the synthesis is the mechanism reliability: in fact, since $\varphi_O < \varphi_r$, there is a certain angle at which the rotary distributor can move without entering the top opening range. By imposing such angles via the precision points, it is ensured that the rocker enters the top opening range in a pre-determined timespan, which can be calculated by choosing the crank rotational speed. In our example, the mechanism is designed to rotate the rocker about 20° from the dead points ($\delta_{3,2} = \delta_{3,1} - \delta_{3,3} = 20°$) when the crank rotates about 45° about its axis ($\delta_{1,2} = \delta_{1,3} - \delta_{1,1} = 45°$). The time needed for such rotation is:

$$t_{span} = \frac{\delta_{1,2}}{\dot{q}} = \frac{\delta_{1,3} - \delta_{1,1}}{\dot{q}} \tag{21}$$

where $\dot{q}$ is the constant rotation speed of the crank.

As a result, the timing of the mechanism is not dependent on the control system, but rather on the mechanical design of the four-bar linkage. The device can be cheaper: there is no need for complex electronics or sensors; instead, the motor can be driven at a constant speed by using a single potentiometer.

**Table 3.** Results of the synthesis of the mechanism. Link lengths to the left, link orientation at configuration 0 to the right.

| Parameter | Value | Unit | Parameter | Value | Unit |
|-----------|-------|------|-----------|-------|------|
| $a_1$ | 39.6 | [mm] | $\varphi_{1,0}$ | −2.55 | [°] |
| $a_2$ | 104.2 | [mm] | $\varphi_{2,0}$ | −13.49 | [°] |
| $a_4$ | 120.6 | [mm] | $\varphi_{4,0}$ | 326.76 | [°] |

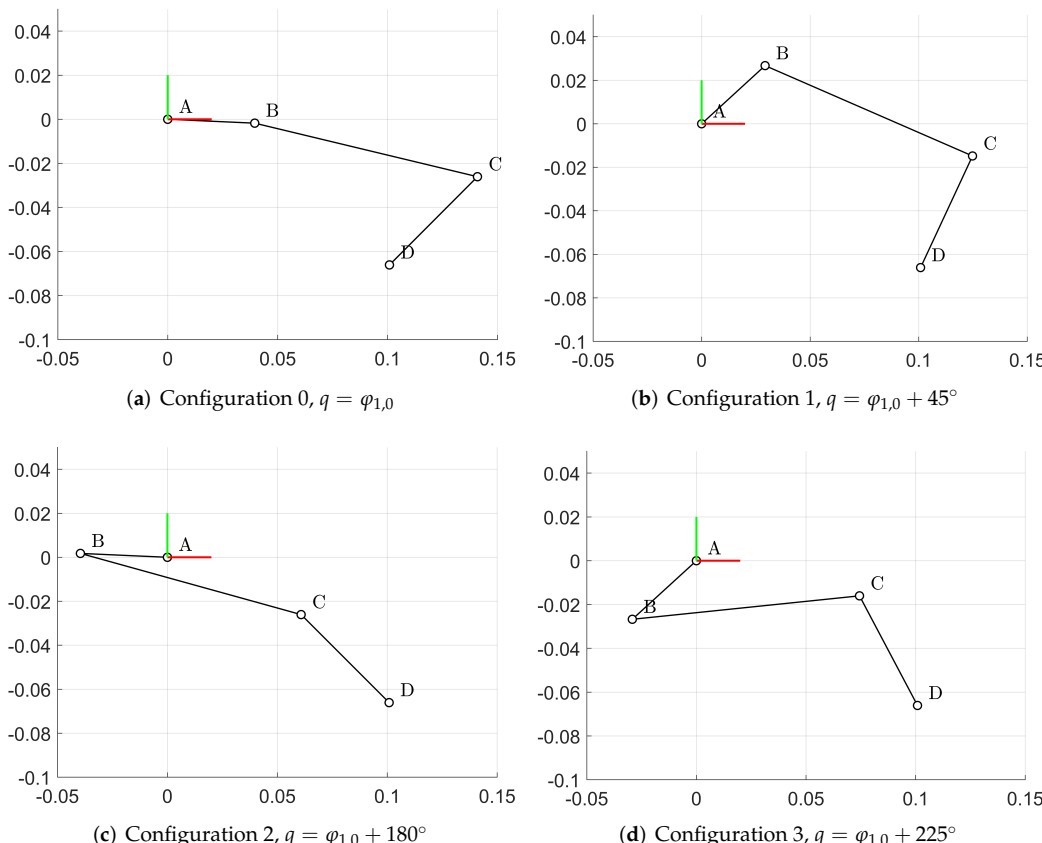

(**a**) Configuration 0, $q = \varphi_{1,0}$

(**b**) Configuration 1, $q = \varphi_{1,0} + 45°$

(**c**) Configuration 2, $q = \varphi_{1,0} + 180°$

(**d**) Configuration 3, $q = \varphi_{1,0} + 225°$

**Figure 4.** Configurations of the synthesis. (**a**) configuration 0; (**b**–**d**) configurations of the precision points. The same configuration numbers are reported in Figure 5.

The overall behavior of the four-bar linkage mechanism for every possible configuration is depicted in Figure 5. As expected, the movement of the rocker can not be controlled precisely between the precision points. In particular, it can be noted how the overall rocker movement range is higher than $\varphi_r$ (in our example, $\max(\varphi_3) - \min(\varphi_3) = 96.32°$), but, actually, every crank half rotation the rocker is placed at exactly 90° with respect to the previous crank half rotation.

*3.2. Blade Design*

From the mechanism synthesis, it is possible to perform the blade design. In this case, the two blades—one for the left compartment, and the one for the right compartment—must be designed separately, since the rocker movement is uneven between the two crank half rotations.

The motion law $\psi(t)$ of Equation (17) is equal to $\varphi_3(q/\dot{q})$ since the motor is moved at constant speed $\dot{q}$, hence there is a direct proportion between the crank rotation $q$ and the movement time $t$. The two motion law for the two blades are retrieved directly from Figure 5a, where:

- for the movement from left to right: $\psi(t) = \varphi_3(t) = \varphi_3(q/\dot{q})$ for $q \in [0, 180]°$;
- for the movement from right to left: $\psi(t) = \varphi_3(t) = \varphi_3(q/\dot{q})$ for $q \in [180, 360]°$.

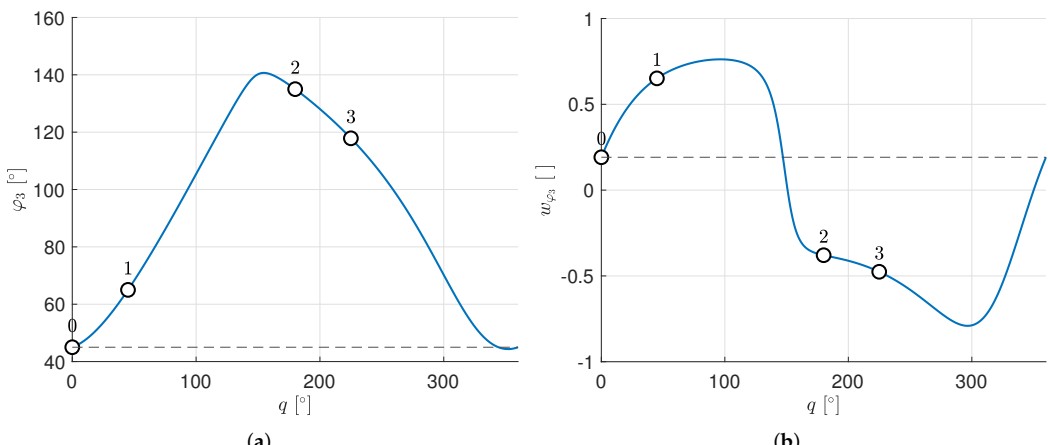

**Figure 5.** Rocker position (**a**) and speed ratio (**b**) for each crank configuration, where $q$ is the displacement with respect to configuration 0. The dots represent the configuration 0 (to the left of each graph) and the three precision points (where $q = \delta_{1,k}$ with $k = 1, 2, 3$). Dashed lines show how $\varphi_3(360°) = \varphi_3(0°)$ and $w_{\varphi_3}(360°) = w_{\varphi_3}(0°)$.

All the numerical values used in the blade design are shown in Table 4, where $\dot{q}$ is calculated by choosing the time required for a crank half rotation $T$ and $r$ is the diameter of the component.

**Table 4.** Numerical values used in the design of the blades.

| Parameter | Left Blade | Right Blade |
|---|---|---|
| $x_0$ [mm] | −8.7 | 8.7 |
| $y_0$ [mm] | −5 | −5 |
| $r$ [mm] | 4 | 4 |
| $T$ [s] | 0.11 | 0.11 |

The two blades are depicted in Figure 6. In light blue (Figure 6a) and orange (Figure 6b) is shown the component relative trajectory during the free fall. Since the blade must be adjacent but not interact with the piece, the top side of the blade is actually the envelope of the bottom part of the component during the fall. Such a side is depicted in the figures with a bold black line.

Please note that the blade, in the simulation, pierces the cylindrical wall. This is due to the fact that the free fall motion allows the component to exit the cylindrical structure in a time $t_{out}$ lower than $T$. As a result, the physical blade should be cut to fit the cylindrical dimensions. This cutting does not influence the component motion: in fact, it is the expression of the fact that the component has yet left the cylinder while the movement of the rotary distributor from one side to the other is still ongoing; thus, $T$ could be further reduced. The value of $t_{out}$ is the lower limit of $T$, and can be calculated by solving Equation (16) for $y(t) = -R$, where $R$ is the radius of the cylinder (Figure 1a):

$$t_{out} = \sqrt{\frac{2(R + y_0)}{g}} \tag{22}$$

which, for our example, with $|R| = a_3$, $t_{out} = 0.1026$ s.

Varying $T$ has an important influence on the final blade shape. In Figure 7 are shown different blades for different $T$ values where, for simplicity, the component trajectories are stopped at $t = t_{out}$. The more the time $T$ increases, the more the shape of the blade resembles a single-curvature arc, and its initial position (the one depicted in Figure 7) tends to align to the vertical direction. In this way, components that fall from the inlet are not blocked; thus, they simply fall through the device without grouping on the blade.

Since the motion law of the blade $\psi(t)$ differs between the left-right and right-left movements (Figure 5a), the blades of the two compartments are different (Figure 5a), although they look very similar. A more intuitive comparison is depicted in Figure 8, where the right blade has been mirrored with respect to the vertical axis. The left blade, which corresponds to the range $q \in [0, 180]°$ presents an evident sharp bend right before the cylinder edges, which corresponds to the last part of the rotary movement, during which the speed rapidly changes (range $q \in [120, 160]°$ of Figure 5b). Such change is less evident in the other movement ($q \in [180, 360]°$), where can be seen a small spike in the rocker speed before returning to configuration 0. Moreover, the first movement presents a generally higher rocker speed, which increases the distance of the component from the horizontal axis, resulting in a blade that bends towards the top of the absolute reference system.

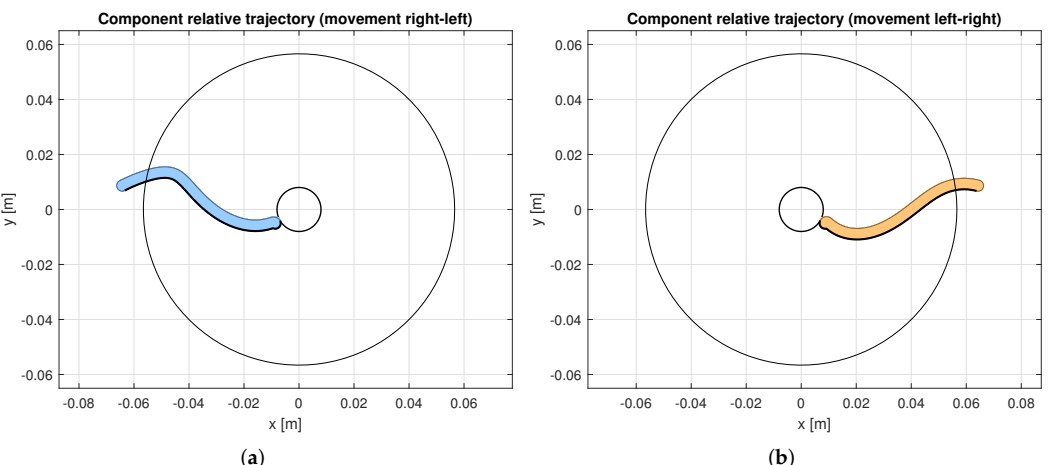

**Figure 6.** Shape of the two blades -left (**a**) and right (**b**)- for the two compartments. Please note that the shape of the blade pierces the cylinder walls since in the movement time $T$ the component is able to fall outside of the cylinder via free fall.

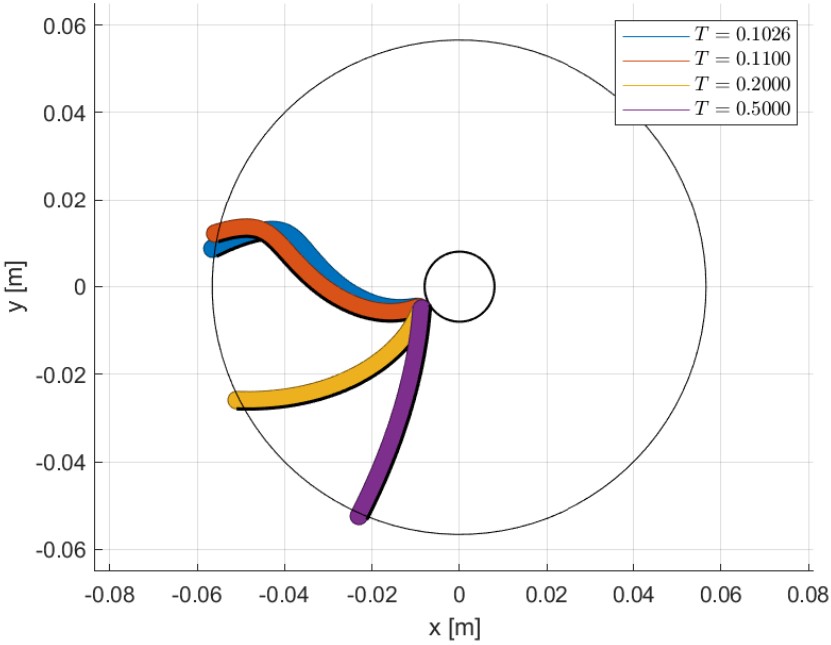

**Figure 7.** Blade shapes for different values of $T$. The trajectories of the components have been stopped evenly at $t = t_{out}$.

From a manufacturing point of view, it is more convenient to produce a single blade rather than two distinct blades. In this sense, it is mandatory, for our example, to consider only the right blade. In fact, if the left blade is considered, during the left-right movement (which would require the right blade) the trajectory of the component in the relative reference frame would still be the orange of Figure 8. As a result, since the trajectory pierces the left blade, the component would interact with the blade, producing unexpected behavior. On the contrary, if the right blade is considered, in the right-left movement the component would fall producing a certain distance to the right blade. Nonetheless, the free fall motion would still be guaranteed.

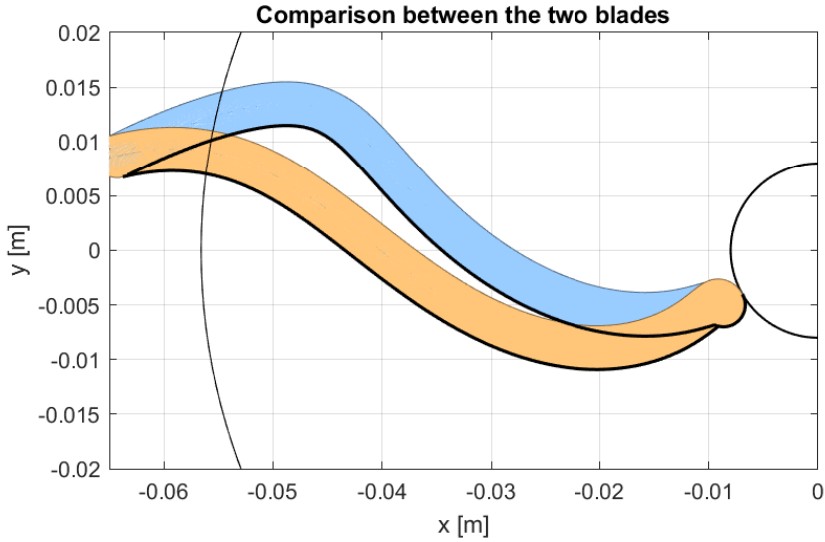

**Figure 8.** Comparison of the two blades.

Finally, it has to be noted that the blade has been designed to be as close to the rotary distributor shaft to be as flexible as possible. In fact, in terms of speed, the initial part of the free falling motion is the slowest, since speed increases linearly with time; thus, the trajectory performed by a component adjacent to the shaft is the worst-case scenario in terms of possible interactions with the blade. Once the blade is designed with the component adjacent to the shaft, any other starting point on the blade ensures no interaction with the blade. An example is provided by Figure 9, in which a different starting point on the blade is considered. In here it can be seen that the component on the left, after a rotation of the blade, has a certain distance from the blade itself; the component whose falling starts in $(x_0, y_0)$ (component to the right), instead, perfectly follows the blade profile. As a result, any component that is placed on any other point on the blade will never interact with the blade and will fall by means of gravity.

*3.3. Motor Torques*

The four-bar linkage comes with a drawback which is related to the variability of the torques that the motor must apply to the crank to move the entire mechanism, especially when it is close to the singular configurations. To this regard, the proposed design has been compared to the direct driven solution shown in [16]. In the previous work, the motion of the blades is directly driven by an electric motor.

To simplify the comparison, only inertial forces are considered, while friction is neglected. Figure 10 represents the motor torques for three different scenarios, the proposed design (blue) and two cases for the direct drive solution, with the motor following a trapezoidal speed law (orange) and a trapezoidal acceleration law (yellow). The speed coefficient $c_v$ is set equal to 1.333; it is defined as

$$c_v = \frac{T}{T - T_a} \tag{23}$$

where $T$ is the motion time for one kitting operation (i.e., 180° for the proposed system and 120° for direct drive solution), and $T_a$ is the acceleration time, which is considered equal to deceleration time. $T$ is set to 0.11 s. As to the trapezoidal acceleration law, the ratio between the time at constant acceleration and the total acceleration time $T_a$, is set equal to 1/3.

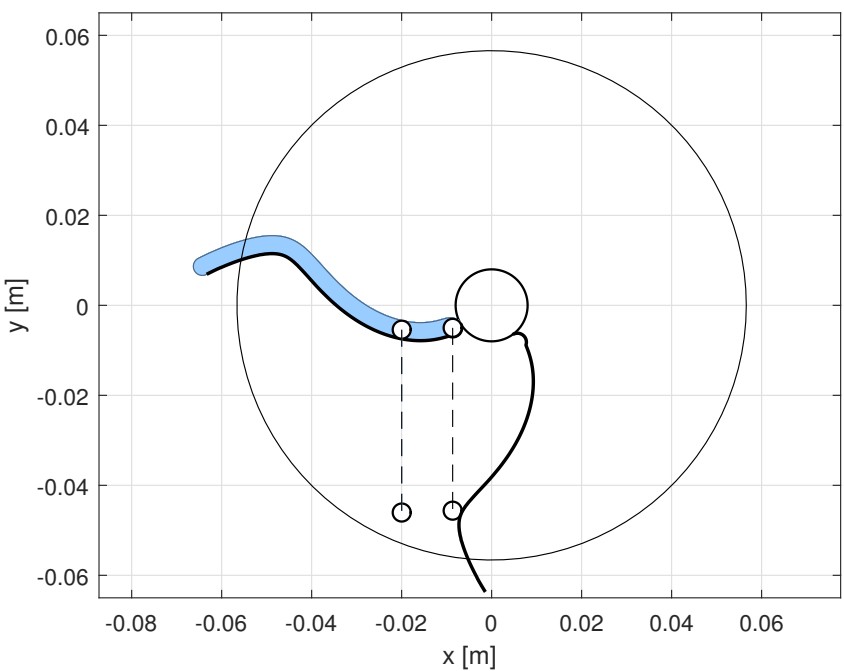

**Figure 9.** Comparison of trajectories of components with different starting points, at the beginning of the blade movement and after a blade rotation of about 140°. A component which starts the fall away from the rotary compartment shaft will not interact with the blade.

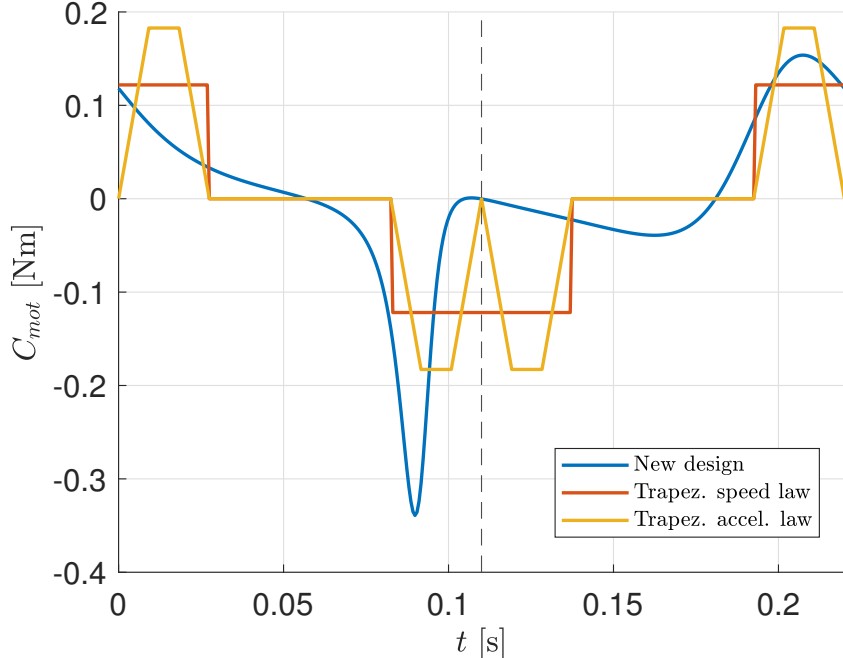

**Figure 10.** Motor torques for different design solutions. Proposed design (blue) and direct motor installed on the rotary distributor: trapezoidal speed law (orange) and trapezoidal acceleration law (yellow).

The proposed system presents a higher value of maximum torque (0.339 Nm) due to the proximity to a singular configuration. However, the root mean square of the motor torque (0.0874 Nm) is comparable to that of the other two solutions (0.0859 Nm for the trapezoidal velocity law and 0.0961 Nm for the trapezoidal acceleration law, respectively), so that they can be regarded as substantially equivalent in terms of torque requirements.

## 4. Experimental Results

To test the effectiveness of the method, an experimental setup has been developed at the University of Padova, as seen in Figure 11.

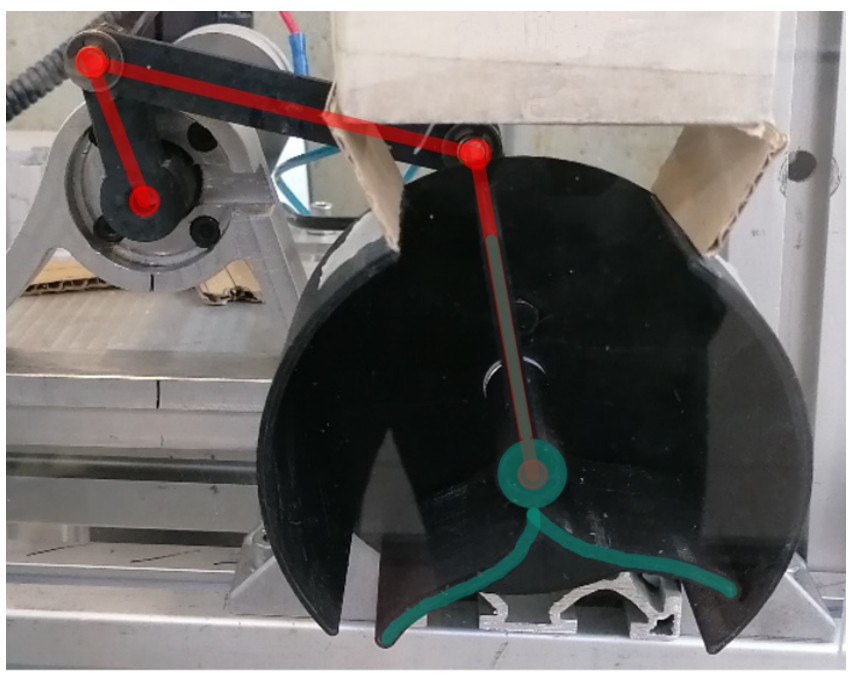

**Figure 11.** Experimental setup. The four-bar linkage mechanism is highlighted in red, while the blades are highlighted in green.

The links, the cylinder, and the blades have been rapidly prototyped by using the plastic material PLA. For the latter, only the right blade has been considered for simplicity. The crank is driven by an MAE M543-0900 motor.

The device has proved its effectiveness with one component and has been tested with multiple components all at once, to simulate the kitting operation, as in Figure 12. Results of the effectiveness of the device with multiple components and different $T$ values are shown in Table 5. The same movement has been performed multiple times to have an empirical information about any possible problems. The components are fed from a grouping station at the top. In Table 5 a check sign ($\checkmark$) indicates that the device, during the tests, have not shown any issues; an approx sign ($\approx$) indicates that only in a minor set of tests ($<5\%$) some issues occurred; finally, a cross sign ($\times$) indicates that with a specific $T$ and number of components the blade is not reliable for industrial use. It has to be noted that, in terms of reliability, only the cases with the check mark can be considered suitable for industrial use.

As expected, the design of the blade is more effective with few components. Indeed, increasing the number of components to be kitted increases the chance of interference between the components; thus, they may not follow the predicted trajectory. With many components the tests have shown pieces stuck between the blades and the cylinder walls, either during the loading or the unloading phase. In particular, with $T = 0.11$ s and 6 components, some pieces have fallen within the wrong compartment.

Please note that most of the issues at low $T$ values occur at inlet side. Optimization of the cylinder walls and/or of the blade dividing the two compartments is likely to reduce

failures, but needs further investigation and testing. Also the influence on failures of the direction and timing of incoming parts is worth to be investigated, focusing on the dynamic interaction between the dividing blade and the falling parts.

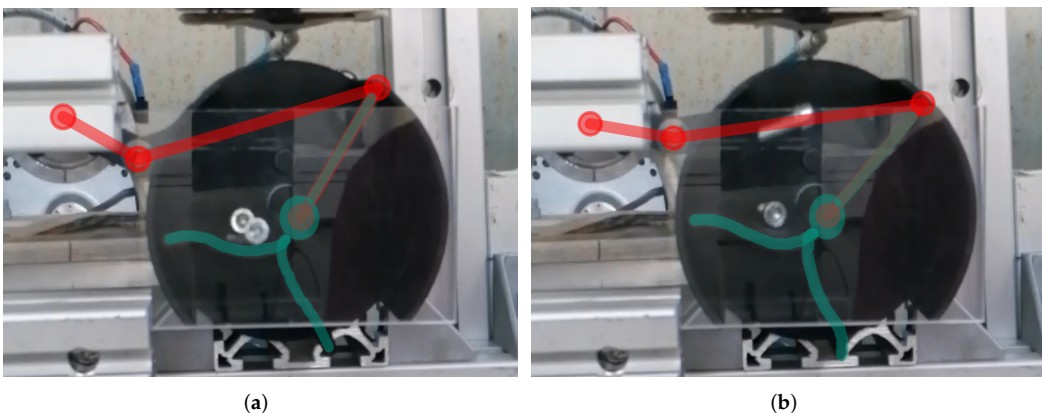

|              |              |
| :----------: | :----------: |
| (**a**)      | (**b**)      |

**Figure 12.** Example of interaction with multiple components. (**a**) the two components are perfectly aligned on the blade before the free fall; (**b**) one component is falling on the other which is starting the free falling motion. The four-bar linkage system is highlighted in red, while the blades are highlighted in green.

**Table 5.** Experimental results of multiple rotations with different $T$ and a different number of components. The blades are proven to be very effective by design with few components, but for small $T$ and a high number of components some problems may occur.

| Components | $T = 0.11$ s | $T = 0.2$ s | $T = 0.5$ s |
| :--------: | :----------: | :---------: | :---------: |
| 1          | ✓            | ✓           | ✓           |
| 2          | ≈            | ✓           | ✓           |
| 6          | ×            | ≈           | ✓           |

## 5. Conclusions

In this paper, the design of a novel feeding device has been proposed. The device is composed of a rotary distributor, divided into two compartments by a linear blade, and a four-bar linkage mechanism used to drive the distributor. Each compartment is used alternatively to group components and let them fall into a following kitting station, while the alternate left-right movements are driven by the kinematics of the four-bar linkage mechanism. The components are accompanied by blades specifically designed to support the components' free-falling motion.

Both the four-bar linkage mechanism and the blade design have been presented in this paper. The mechanism design is performed by a specific functional synthesis, which requires three parameters to be performed algebraically. The blade shape, on the other hand, is designed exploiting the four-bar linkage kinematic results.

The main advantage of the proposed mechanism is its simplicity in the control system. In fact, the movement of the blade is driven solely by the kinematics of the four-bar linkage, where the crank is fixed to the shaft of an electric motor controlled at a fixed speed, thus requiring a very simple control system.

The device has proved to be reliable for very few components, whereas multiple pieces can interact with each other, resulting in unexpected behavior, especially at high speed. Nonetheless, the fact that loading and unloading of the compartments can be performed in a single movement—since while one compartment is unloading the other can be loaded—shows very promising expectations for future developments.

**Author Contributions:** Conceptualization, M.B. and G.R.; methodology, G.R.; software, M.B. and R.M.; validation, M.B. and R.M.; formal analysis, M.B. and G.R.; investigation, R.M.; resources, G.R.; writing—original draft preparation, M.B. and R.M.; writing—review and editing, R.M. and G.R.; visualization, M.B. and R.M.; supervision, G.R. All authors have read and agreed to the published version of the manuscript.

**Funding:** This research received no external funding.

**Institutional Review Board Statement:** Not applicable.

**Informed Consent Statement:** Not applicable.

**Data Availability Statement:** Not applicable.

**Conflicts of Interest:** The authors declare no conflict of interest.

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
