# Peer review of "Design of the Drive Mechanism of a Rotating Feeding Device"

_machines, doi:10.3390/machines10121160_

Round 1

Reviewer 1 Report

An important topic from the production point of view. Manufacturers of many companies are interested in reducing production costs and increasing efficiency. It is important to take into account the smallest and the greatest deflection of the arm, as this can cause the most problems. Does the speed of getting the parts inside depend / can be dependent on the rotational speed? Will the device get stuck at the inlet and outlet of large components? Shouldn't the shape of the blades change when the rotational speed is changed? High rotational speeds may require additional movement of incoming parts rebounding from the blades. What materials will the blades be made of? Different materials spring back and reflect falling parts differently.

Author Response

We thank the reviewer for the constructive comments. Please find attached the corresponding answers.

Reviewer 2 Report

In this investigation, the authors proposed a new rotary device, which is driven by a four-bar linkage mechanism. In general, the work is worth investigating. However, there are some issues need to be addressed.

1) The authors state that the “four-bar linkage mechanism” provides some benefits, such as control on the displacements of the rotating device without requiring complex electronics. Compared with conventional system, however, the "four-bar linkage mechanism" can create a design with a too-high torque situation, which may lead to the failure of the structure. In addition, multiple pieces may interfere with each other. I highly recommend the authors to consider these problems and provide corresponding solutions.

2) The authors are suggested to provide rationales behind their design.

3) Please enlarge font size in Figures 4 and 5.

4) I could not see the structure in Figures 10 and 11. Please improve these figures.

Author Response

(The authors gave the same response as above.)

Reviewer 3 Report

The article is devoted to the improvement of the feeding device. Motion of the rotator is driven by a four-bar link mechanism developed through functional synthesis. The rotary distributor's alternating movement comes from a constant motor speed. The research topic is quite relevant.

The work presents the mechanical design of the mechanism and blades, contains a theoretical substantiation of the developed scheme and a description of the created prototype.

The paper is qualified and well structured.

There are the following comments:

1. The description of the operation of the mechanism needs to be improved. In particular, in order to improve understanding of the principles of operation of the device, the authors are invited to change Fig. 2 (for example, by adding the designation of the entry and exit points of the components and explicitly indicating the direction of movement of the components) and add a description of this figure to the text.

2. Chapter 4 should be expanded by adding an analysis of the results obtained from experiments.

3. The quality of the pic. 11 should be improved

4. The list of references contains 4 works of the authors of the article. It is proposed to expand the number of references or reduce the number of references to oneself.

Small remarks:

1. Some sentences are difficult to understand.

For example, in paragraph 2.2, page 7:

“Indeed, it is nearly impossible to drive the distributor at a constant speed, since it would allow a very short time for the components to fall within the compartments. ”

2. It is necessary to monitor the correct use of terms and avoid speech turns.

For example, in paragraph 3, page 10:

“As long as T increases, the blade BECOMES MORE VERTICAL”

"IDEALLY, if T → ∞, the blade becomes vertical."

Page eleven:

"...the trajectory of the component in the relative reference frame would STILL BE THE ORANGE of Figure 8."

1.       There are typos in the text:

Page 5

“by a very simple control system. I

Page 4

"Table 1 shows the number of unknown parameters"

Page 5

"1 for configuration 0. please note that by"

Page eight

"The two motion law for the two blades are retrieved"

Page 12

"...any other starting point on the blade ensure no interaction with the blade"

"...to have an empirical information.."

Page 13

“…noted that for an industrial-grade reliability…”

Author Response

(The authors gave the same response as above.)

Round 2

Reviewer 2 Report

Please clearly indicate how you address all the issues point by point.

Reviewer 3 Report

The authors did a good job of improving the article.

The only remark is concerned with misprint in page 13:

"have an empirical information".

The word "information" is an uncountable noun.